# Affordable Broadband with Software Defined IPv6 Network for Developing Rural Communities

**Babu R. Dawadi [1,*], Danda B. Rawat [2,*] , Shashidhar R. Joshi [1] and Daya S. Baral [1]**

[1]  Depart of Electronics and Computer Engineering, Pulchowk Campus, Tribhuvan University, Kathmandu 19758, Nepal; srjoshi@ioe.edu.np (S.R.J.); dsbaral@ioe.edu.np (D.S.B.)

[2]  Cyber Security and Wireless Networking Innovations Lab, EECS Department, Howard University, Washington, DC 20059, USA

*   Correspondence: baburd@ioe.edu.np (B.R.D.); db.rawat@ieee.org (D.B.R.)

**Abstract:** The software defined networking (SDN) paradigm with enhanced features of IPv6 offers flexible network management and better network visibility for enhancing overall network performance, network manageability, and security. Thus, along with the IPv6 network deployment worldwide, SDN migration has emerged worldwide, but network service providers suffer from different issues when migrating their existing legacy network into operable SDN and IPv6 enabled networks. In this paper, we investigate the affordability of broadband network services for the rural communities in the context of information and communication technology (ICT) infrastructure deployment throughout Nepal. During the phase of network transformation, it will be more challenging for the service providers of Nepal to have a proper choice of technologies to expand the network while considering the proper policy formulation, affordability, need of skilled human resources, deployment cost, and many other aspects. We also present the service provider affordability via energy optimization in software defined IPv6 network (SoDIP6) implementation that contributes to a reduction in organizational operational expenditure (OpEX). We perform an experimental analysis over an SoDIP6 network testbed and present a comparison of the annual energy and OpEX savings for network service providers. Our empirical analysis shows that an energy saving of 31.50% on switches and 55.44% on links can be achieved with an SoDIP6 network compared to a network with legacy devices and network management. Optimization on service provider network operational cost leads to sustainability and affordable services to both customers and service providers

**Keywords:** affordable broadband; rural communities; sustainable societies; SDN; IPv6; SoDIP6; energy efficiency

## 1. Introduction

The world is implementing information and communication technology (ICT) infrastructure and services at an unprecedented rate. The over 60% overall broadband internet penetration rate and 12% fixed broadband rate (as of March 2019) of Nepal [1] indicates that there is still sufficient space for Nepalese network service providers to expand their services throughout the country, while the government has to pave the way through effective policy to enhance ICT penetration across the country. Less than 1% fixed broadband penetration in rural Nepal indicates that the majority of rural area in Nepal is without network coverage and connectivity. Most rural areas of developing countries like Nepal have limited resources like public libraries, skilled manpower, infrastructure, etc. [2]. Recently, the government regulatory body of Nepal has proposed a progressive plan to expand ICT infrastructure throughout the country. This is considered as the next milestone for Nepal to

improve people's life style with the proper use of ICT in different sectors like healthcare, agriculture, industry, governance, and many more.

Almost 49% of the total population of Nepal live in the hilly zone including at high altitude, which covers almost 70% of total land. The major issues for rural communities are the diverse geography and demography leading to lack of accessibility, literacy, affordability, and sustainability. Although ICT can play a remarkable role in uplifting people's living status, the prerequisites to provide broadband service as basic service in rural Nepal are reliable internet connectivity, reliable power supply, affordable education to improve literacy rate, efficient management and delivery of local products, encouragement of private sectors to expand services with the latest technologies to rural communities, effective policies and implementation, subsidization for infrastructure expansion, content delivery, and quality of services for the users.

In this paper, we discuss different ways to provide affordable broadband services to rural communities in Nepal. Particularly, we discuss the factors that make the broadband services more affordable and sustainable for rural societies. Then, we present the operational cost optimization and efficiency for service providers by implementing the proposed software defined IPv6 (SoDIP6) networks so as to provide affordable network services to underserved people and optimize sustainability for network service providers.

The remainder of this paper is structured as follows. Section 2 presents policy initiatives and network deployment activities of Nepal for nation-wide broadband expansion. Different approaches with recommendations to be adopted by stakeholders to provide affordable broadband services are discussed in Section 3. Section 4 presents the experimental setup and analysis of the proposed SoDIP6 network in terms of power consumption optimization to measure the affordability of broadband services to communities. Section 5 concludes the paper.

## 2. Broadband Policy and Network Deployment Initiatives of Nepal

Nepal has adapted a very liberal policy in ICT sector development with the promulgation of the Telecommunication Act 2003 and the Telecom Policy 2004. The policy and act encourage involvement of private sectors in ICT. Private industries are being attracted in the ICT sector with sufficient investment for ICT service provisioning. This helps reduce the digital divide and considerably increase the ICT penetration rate. However, the increase in penetration rate is not contributing to rural communities, but more services are available in urban areas. Because of the sparse deployment of ICT infrastructure and their services to rural areas, connectivity is very poor. However, ICT infrastructure is perceived as the lynch-pin for sustainable economies of rural communities [3].

The complex territory of Nepal by its geography has three main divided regions: north—the section known as the 'Mountainous Region' (altitude between 4877 and 8848 m), mid—the section known as the 'Hilly Region' (altitude between 610 to 4876 m), and south—the section called the 'Terai Region' (regions below elevation 610 m). Nepal is a mountainous country because about 77% of total land is covered by mountains and hills. By demography, the mountain and hilly zone cover only 49.7% of the population while the Terai zone has 51.3% population in about 23% of the total land area. More developed cities are located in the Terai region, leading to increasing population migration day-by-day from mountain/hill areas. According to [4], the projected population of Nepal in 2016 was 28.4 million, in which the urban population was almost 20 million and the rural population was 8.4 million. In this context, rural Nepal is basically the hilly area and lacks in reliable infrastructure such as road networks, electricity, ICT networks, etc., leading to a high cost to expand ICT services in rural Nepal. Currently about 40 internet service providers (ISPs) are providing internet and network services, mostly focusing their services in the urban areas. Similarly, six telecom operators (Telcos) are providing voice and data services. However, existing urban centric networks and services are almost sufficient to achieve the targets set by the ICT and broadband policies of Nepal for urban communities, to achieve targets accordingly for rural communities, requires a rapid pace of ICT and broadband infrastructure expansion [5].

The history of internet penetration shows that overall penetration of Nepal will cross 100% after the year 2023, including mobile GPRS (General Packet Radio Service) users. On the other hand, the ICT Policy-2015 [6] and Broadband Policy-2015 [7] have put 2020 targets regarding network expansion and ICT enhancement throughout the country. For example, (i) Nepal will be at least in the top second quantile of the international ICT development index and e-Government rankings by 2020; (ii) at least 75% of the population will have digital literacy skills by the end of 2020; (iii) 90% of the Nepalese population will be able to access the broadband services by 2020; (iv) the GDP growth accounting by ICT will be at least 7.5% by 2020; (v) by 2020, the entire population of Nepal will have access to the internet; (vi) 80% of all citizens facing government services will be offered online services by 2020 [5].

A nation-wide broadband deployment project including district level optical fiber network connectivity through the Mid-Hill Highway; establishment/erection of mobile base transceiver stations, including operation and maintenance of shelter and power; connecting schools and connecting communities; and making ICT accessible to people with disabilities are the major initiative by Nepal Telecommunications Authority (NTA) utilizing the Rural Telecommunication Disbursement Fund (RTDF) that supports the policy actions irrespective of the time of target defined. With the deployment of nation-wide ICT infrastructure, the government has to come up with additional measures to reduce the digital divide and gap in internet access and utilization between groups [8]. Hence, the increase in ICT accessibility to rural areas becomes the basic requirement that these projects are trying to address. Additionally, content creation, delivery, and security will become other concerns, together with the greening of network infrastructure for the ecofriendly environment and long term sustainability of service providers and societies [5].

## 3. Affordable Broadband for Rural Communities

This section presents a study about "How to make broadband services affordable for rural communities in the context of Nepal?" We present the common approaches where broadband connectivity and services to rural Nepal are more cost effective and sustainable. Then, we evaluate the energy efficiency of SoDIP6 network to verify that it has better energy saving than legacy IPv4 networking system and hence reduces the organizational capital and operational expenditures (CapEX/OpEX) that contribute to service providers' sustainability.

There are some factors that affect the affordability and sustainability of rural communities in the deployment of rural broadband infrastructure and services [2,9]. These are: (i) low household income; (ii) lower literacy rate; (iii) sparse population; (iv) difficult terrain; (v) higher rate of population migration; (vi) energy; (vii) accessibility; (viii) policy and regulatory environment; (viii) broadband price; (ix) technology deployment cost; (x) technical human resources; (xi) local barriers; (xii) content and language barriers.

Briefly summarizing the above points, it is realized that rural people have low purchasing power, and severely lack the opportunities to get efficient government services, transportation, education, healthcare, etc. [10]. In the last two decades in Nepal, people form the hilly areas have migrated to Terai areas due to the limited resources, as well as other seen/unseen political movements. Energy is the major pillar of economic development of the country, but the lack of a reliable electricity supply system in rural Nepal, lack of connectivity such as road networks to make the rural area accessible, higher network deployment cost, absence of governance and citizen services for the rural communities, lack of dedicated and visionary policy are the major hindrances to having affordable and sustainable broadband services. Similarly, road network and broadband infrastructure expansion is still difficult due to unnecessary obstructions by local people, direct/indirect political movements with language barriers, in which more than a hundred languages are spoken in different parts of the country. In this context, NTA initiations to improve the availability of ICT infrastructure only, do not guarantee the improvement of living standards of rural communities. However, the use of ICT stimulates activities to automate different areas such as local industries, agriculture, healthcare, education, etc., the primary concern is the strong presence of government regarding connectivity and coverage, then secondarily

ICT and awareness in the unserved and underserved areas of rural Nepal. The existing national ICT and broadband policies [6,7] seem to be amended for dedication, suitable vision, and strategies to attract service providers to offer high-speed network connections and services for rural communities.

Table 1 presents the price (in USD) of fixed internet broadband services provided by major three ISPs (World Link, Vianet, and Subishu Cable Net Pvt. Ltd., Kathmandu, Nepal) of Nepal. Those prices are considerably affordable for the city/urban communities but not affordable for the rural communities of Nepal. Similarly, Table 2 shows that Nepal has the third highest prepaid mobile broadband prices in South Asia.

**Table 1.** Fix broadband prices of major ISPs of Nepal (As of March 2019, www.worldlink.con.np (accessed on 22 June 2019), www.vianet.com.np (accessed on 22 June 2019), https://subisu.net.np/ (accessed on 22 June 2019)).

| Bandwidth (Mbps) | Annual Price (USD) (Assuming 1 USD = 110 NRS) | | |
| :---: | :---: | :---: | :---: |
| | World Link | Vianet | Subishu |
| 15 | - | - | 104.45 |
| 20 | - | 130.90 | 126.27 |
| 25 | 136.36 | - | - |
| 30 | - | 152.72 | 159 |
| 40 | 159.09 | - | - |
| 60 | 236.36 | 187.63 | 202.63 |

**Table 2.** Mobile broadband prices in South Asian countries [11].

| Country | Service Provider | Price of Plan (Local Currency, incl. tax) | Validity (days) | Tax % | Data Cap (GB) | Basket in Local Currency | Price (USD) |
| :---: | :---: | :---: | :---: | :---: | :---: | :---: | :---: |
| Afghanistan | Roshan | 350 | 30 | 0 | 2 | 350 | 4.65 |
| Bangladesh | Grameen Phone | 229 | 30 | 21 | 1.5 | 229 | 2.77 |
| Bhutan | Bhutan Telecom | 209.00 | 30.00 | 5 | 1.7 | 209 | 2.95 |
| India | Reliance Jio | 149 | 28 | 18 | 42 | 149 | 2.10 |
| Maldives | Dhiraagu | 230 | 30 | 6 | 2.5 | 230 | 15.92 |
| Nepal | Ncell | 113.64 | 7 | 26 | 1 | 454.56 | 4.03 |
| Pakistan | Jazz | 110 | 7 | 0 | 1.9 | 440 | 3.17 |
| Sri Lanka | Dialog | 349.00 | 30 | 19.74 | 2 | 349 | 1.93 |

Based on the literature survey, to have affordable broadband services for rural communities, we present main focus areas as depicted in Figure 1. These are basically categorized into (i) policy, (ii) infrastructure, and (iii) innovation, research, and development. The subtopics listed under these focus areas are briefly summarized here.

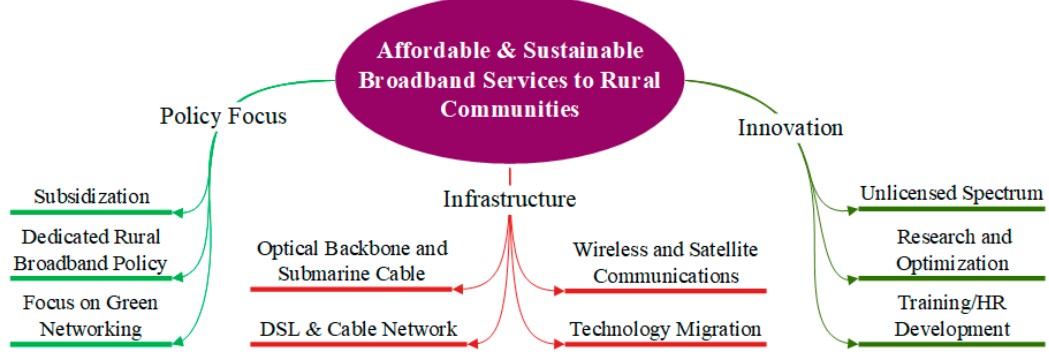

**Figure 1.** Focus areas for affordable and sustainable broadband services. DLS, digital subscriber line.

### 3.1. Subsidization

Even if the network infrastructure is ready for rural communities, service providers will not be attracted to provide services to rural areas due to a lack of guarantee of return on investment. Hence government grants and loan in the form of subsidization are to be provided to run their digital infrastructure under the community network. These subsidies could be used to operate health centers, hospitals, schools, community farms, etc., so that they will be encouraged to use rural utilities and other broadband services [12]. Access to government information to be delivered to citizens like .np domain content, educational materials, health and agricultural farming related information should be available free of cost to rural communities in the beginning so that rural people will be attracted to ICT based services. For example, fully subsidized IT lab setup at government schools throughout Nepal is a good initiation by the NTA that helps to establish IT based education centers in rural areas of Nepal [13]. Free access to content can also be ensured via a geographical limit in which marginalized citizens of those areas can be set as a priority to provide free access to internet and services. Digital content production in their own language and sharing among the rural communities can stimulate societal development from within the societies [2].

Developed countries like America and South Korea had put direct investment as well as guarantee loans into the rural communities to reduce the digital divide via reliable broadband infrastructure and services [14]. Discounts or direct funding to end users as incentive based subsidies may be more effective via the provision of broadband vouchers to economically marginalized people in rural areas [15].

Service providers will be encouraged to expand their infrastructure and services to rural areas only if they are guaranteed some subsidies like tax exemptions on the network equipment that is imported for rural broadband expansion. Hence, exemption on local taxes and or taxes while importing equipment as well as direct subsidies through public private partnerships (PPP) can be considered as an important avenue for social development and reduce the digital divide by means of reliable and sustainable broadband services [16]. As a priority, subsidies need to be guaranteed first for building a new network rather than upgrading the existing network [17].

### 3.2. Dedicated Rural Broadband Policy

In particular, considering the rural areas of Nepal, adequate policy intervention is required to develop a sustainable, digital, rural society. A dedicated and community focused policy helps to address the subsidization schemes mentioned in the above section. The recent broadband policy of Nepal [7] requires additional amendment for the effective mobilization of the RTDF to provide subsidies to communities and service providers to run the broadband infrastructure and services in a PPP model. Clear policy vision is to be derived to address the sustainability issues of disadvantaged schools, public libraries, community health centers, and many more [3]. For example, low income households with their children enrolled in community schools can be provided broadband internet for as little as possible or free of cost [8]. Broadband services will have significant impact on the economic growth of rural communities in their organizations and small businesses [18]. With a favorable environment created by the government, if guaranteed via policy means, affordable services and operational sustainability can be achieved with broadband networks in rural and remote areas [19].

### 3.3. Focus on Green Networking

With the exponential growth of internet users worldwide, evolving smart cities with an increase of wireless sensor networks and internet-of-things devices, the energy consumption by ICT equipment is rapidly increasing, leading to increased carbon dioxide ($CO_2$) emission. The volume of $CO_2$ emission produced by the ICT sector is estimated to be at least 2% of the world's total carbon footprint in 2020 [20]. The ICT sector is forecasted to contribute 1.3 Gt of greenhouse gas emissions in 2020 [21]. Global warming and the effects of climate change are more serious in the context of Nepal because it is

a mountainous country. The rapidly retreating glaciers (average retreat rate of more than 30 M/year), erratic rainfall and frequent flooding, and severe drought leads to alarming situations for a developing country such as Nepal [22]. With the increasing network connectivity to rural Nepal covering the hilly areas, the effect of global warming will be alarming. With the increasing ICT infrastructure and users, the sector consumes more energy leading to a higher volume of $CO_2$ emission, and hence this scenario is expected to increase rapidly if no mitigation approaches via greening techniques are adopted. Greening of networks such as the use of renewable energy technologies to power the network equipment and the use of the latest energy efficient networking technologies like software defined networking (SDN) enabled IPv6 networks while deploying the connection would help to create sustainable societies, as well as to provide support to service providers' sustainability. Significant amounts of energy savings in an ISP's network would help to reduce the annual energy bill as a part of organizational OpEX [23]. Considering the greening approach by newer networking technologies like SDN and IPv6, we can find that there are several studies that contribute to green ICT with an energy aware routing scheme for energy saving and carbon footprint reduction to show that IPv6 and SDN have greater flexibility towards green networking [20,24–27].

### 3.4. Optical Backbone and Submarine Cable

With the reducing price of optical fiber cable and the requirement of high-speed broadband networks, fiber cable becomes the default choice to establish a nation-wide broadband network. RTDF funded project as initiated by the NTA will establish a fiber backbone connecting every district head quarter by 96-core fiber and every rural municipality wherever possible by a 48-core fiber backbone. Due to powerful wind at high altitudes and frequent snow fall with zig-zag locations in rural areas of Nepal, in most places, underground backbone fiber installation is not viable. This issue can be avoided if we deploy optical submarine cables recognized as fiber packed armored cable (F-PAC) in Nepalese rural areas. As per the proposal of Murata et al. [28], their practical implementations indicate that F-PAC cabling can have aerial laying and under-ground laying with sufficient counter measures against birds and animal attacks. Hence, it is more cost effective with better safety, security, and reliability than deploying wireless and satellite links for broadband comminutions [29].

### 3.5. Digital Subscriber Line and Cable Network

Fiber to the home (FTTH) has become more popular worldwide to provide broadband communication services [30]. However, in the Nepalese context, FTTH service has just entered into the market as a major competitor, but Digital Subscriber Line (DSL), basically an Asymmetric DSL (ADSL) service, and cable-based services are still recognized as popular connectivity services that are implemented by service providers. In most of the offices where public switched telephone service is available, Nepal Telecom (NT), the incumbent operator, has provided ADSL connectivity throughout Nepal. ADSL became the proprietary of the incumbent operator in Nepal, hence internet and cable TV services through coax connection is also a popular service adapted by ISPs and cable service providers, but they are mostly city/urban centric. Wireless internet connectivity services provided by the Nepal Wireless project (http://www.nepalwireless.net/ (accessed on 24 August 2019), http://www.nepalwireless.com.np/ (accessed on 24 August 2019)) in some of the rural areas of Nepal and NT's ADSL services to every rural municipality, including 3G/GPRS and WiMAX services [31], are the basic connectivity available in rural areas. Based on the viability and possibility of network expansion, local wireless services utilizing TV White Space (TVWS) also enabled low cost and affordable services for rural communities.

### 3.6. Wireless and Satellite Communications

In the mountainous regions where a wired network is not possible, wireless communication such as Wi-MAX and 3G/LTE/4G technologies and the use of TVWS for long distance wireless coverage has to be established. Additionally, some places require satellite communications. Geostationary Earth Orbit satellites are comparatively simple and cost effective compared with other constellations [17]. Focusing on the unserved and underserved communities, Low Earth Orbit (LEO) satellites provide low latency and high-speed broadband services to rural communities. The LEO satellite system is becoming the global communication service provider's counterpart to local terrestrial paging, cellular, and fiber networks. Telesat, OneWeb, SpaceX [32,33], and Amazon [34] have initiated LEO satellites to establish low cost, affordable, and sustainable global communications.

### 3.7. Technology Migration

With the rapidly changing global scenarios with the latest cutting-edge technologies such as advancement in hardware, software, applications, and protocols, service providers must be well adapted to provide efficient, reliable, and fast services to customers. The latest technologies developed will be more robust, efficient, cost effective, and more operation friendly. Hence, encouragement to service providers towards timely migration to newer wired and wireless technologies like SDN enabled IPv6 networks, wireless virtualization, and the use of OpenBTS and OpenCellular technologies are required for sustainable services. However, SDN and IPv6 network migration is not in the scope of this paper, the benefit of its implementation as an energy efficient technology is presented in Section 4.

### 3.8. Unlicensed Spectrum

The government of Nepal has already announced that all cable television companies must transform their cable service network from analogue to digital television broadcasting. In this context, not all channels of the ultra-high frequency TV band are occupied at each location. The portion of the spectrum left unused by broadcasting, known as TV white spaces (TVWS) that have traditionally been used for analogue TV broadcasting and its opening for cognitive access, has greater opportunity to avoid spectrum scarcity issues [35,36]. Radio signals with TVWS are suitable for travelling long distances over hills, mountains, and around/through buildings. This is more applicable in the context of rural Nepal, where in most places it is not possible to connect via wired network. Most of the countries worldwide are exploring the ways to use TVWS for wireless broadband communications [17,35,36]. Hence, low power telecommunications transmissions leading to affordable broadband for rural communities, can also be achieved with the use of TVWS.

### 3.9. Research and Optimization

Innovations, research, development and service optimization are the major entities to develop cost effective and provide sustainable services based on the end users or community requirements, demographic and geographic situation, costumes, cultures, and many other aspects of the rural communities. The actual needs of the society have to be analyzed, and suitable technology has to be implemented. Current RTDF utilization to develop a broadband network as initiated by the NTA is expected to uplift the people's economic status, including improvements in several other dimensions of society. For this, we need suitable policies, plans, and strategies so that government services can be effectively provided to the citizens, including promotions of local industries and the creation of an environment for effective use of technologies to develop sustainable rural societies. Hence, broadband data collections from rural communities to effectively analyze the existing services, customer needs, and requirements will help to get back into action after research to optimize the service use and cost of operation.

*3.10. Training and Human Resource Development*

The sustainability issue is raised mainly due to the lack of skilled human resources to maintain and operate network systems in order to provide broadband services in rural areas. Similarly, the lack of knowledge of consumers about access and use of the internet, including other factors like socioeconomic and demographic factors, are regarded as the challenges of sustainable broadband services for rural communities. Hence, public and private stakeholders in the rural communities are to be trained through consumer education with training initiatives and a broadband affordability program, including other incentives to attract people to use the services [37].

## 4. Affordable Broadband Services with SoDIP6 Network

This section addresses the research question of "How does SDN and IPv6 network implementation contribute to service provider sustainability and affordable broadband services for customers?"

We evaluate the energy optimization in an SoDIP6 network from the experimental analysis by simulation, considering heterogeneous (wired and wireless) networks suitable to be deployed in rural communities. First, we summarize the features of the SoDIP6 network, then we present the energy evaluation of the SoDIP6 network with results and analysis.

SDN and IPv6 are the two networking paradigms that will likely replace the existing legacy IPv4 networking system worldwide to avoid existing issues in addressing, routing, network operation, and management. The major benefit of SDN is its flexibility, programmability, and controllability, which could help to reduce the CapEX/OpEX of organizations [38] and encourages service providers to search for better options and attract them towards SDN. In summary, the benefits of transitioning to an SoDIP6 network are summarized as follows:

- **Address space:** The larger IPv6 address space opens opportunities for scalable network design, growth of end users and smart devices. The existing Network Address Translation (NAT) solution and its associated problems with IPv4 networking will be avoided.
- **Network operation cost:** A flexible and horizontally integrated programmable SoDIP6 network decreases the overall operation and administration cost. Removal of NAT reduces the network operation cost and saves significant annual revenue with increased efficiency of the network in a large scale.
- **Quality of service (QoS):** While there is limited implementation in practice, the flow-level field of the IPv6 header provides spaces for researchers to introduce QoS aware networking operations and services [39–42]. Better quality increases the reliability, and so encourages organizations to move to digital packet-based communication, such as voice over internet protocol, that may lead to more savings in the annual telephony expenditure. Similarly, the programmability and flexibility of SDN would help improve the quality of network services.
- **Security:** Security threats and worldwide cybercrime cost people and businesses, a loss of more than 200 billion USD per year [43]. The built-in IP Security (IPSec) framework in IPv6 implementation can significantly improve the network security [44–46]. Similarly, programmability features in SDN enables the application of different security models and deals with several security challenges to avoid modern security breaches [47–53].

We consider an end-access ISP network consisting of wired and wireless customer premise equipment (CPE), switches and their associated links by developing a custom topology in a Mininet-WiFi emulator with an Open Daylight SDN controller over an Ubuntu virtual machine. Considering Figure 2, the end-access network newly deployed is enabled with SDN and IPv6. The SoDIP6 enabled end-access network topology consists of 14 switches (SDN switch: S0–S5, Edge Switch: E1–E5 and Wireless Access Point: A1–A3) and 8 CPEs (cpe1–cpe5 and sta1–sta3) including wireless stations. The end access gateway router is supposed to be a translator router that performs the translation of SoDIP6 traffic to Legacy IPv4 and vice versa. We also assumed that the ISP backbone network is not SoDIP6 capable.

This means it has only legacy IPv4 supports. Hence, end access gateway can be treated as a customer side translator and the external gateway as a provider side translator. These are translator routers which perform IPv4 to IPv6 header translations and vice versa.

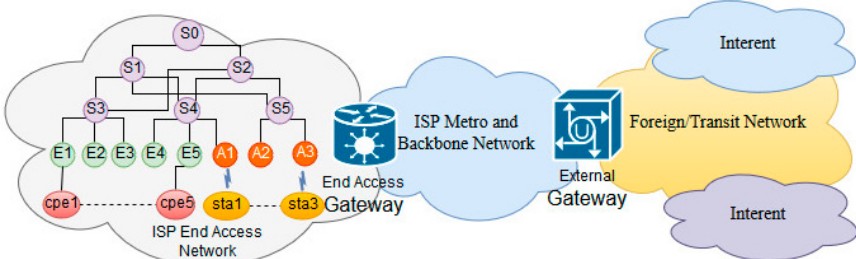

**Figure 2.** End access network in the ISP (Internet Service Provider) network infrastructure.

Legacy IPv4 networking devices like switches and links run 24 h a day with peak energy consumptions, and so result in a higher energy bill annually. The beauty of an SoDIP6 network is that energy consumption can be optimized using dynamic scaling, adaptive link rate, and smart sleeping [54]. We consider smart sleeping and traffic load-based energy variance in this experimental analysis to evaluate the energy efficiency of the proposed testbed. The overall steps of experimental activities are depicted in Figure 3. Custom end-access network topology as mentioned in Figure 2 was loaded and executed in the Mininet emulator after starting the Open Daylight SDN controller. The network hosts generated random IPv6 traffic in a periodic manner. The traffic with power consumption status of data plane devices was continuously monitored via a power monitoring daemon. Similarly, the sleep/wakeup daemon instructed data plane devices based on traffic status to sleep or wakeup accordingly. The pattern of energy consumption recording during 24 h of monitoring is presented in Algorithm 1. In the running of the SoDIP6 network, at the beginning, a list of source hosts and destination hosts was randomly generated to establish IPv6 traffic exchange using a traffic generator tool like 'iperf' and 'ping6'. Source hosts periodically generated IPv6 traffic to the destination, while the iperf tool generated traffic every 2 min and the ping6 tool generated traffic every 3 min. This situation created continuous traffic flow in the network. The device and link power monitoring daemon continuously monitored the IPv6 traffic status at the switch and links then activated the smart sleeping and wakeup daemon accordingly.

We also consider that a switch has 110 Watt power consumption in normal conditions [55] and full load links consumes 40 Watt [56]. Similarly, per port power consumption by switch ports per Mbps was 0.01 Watt [57,58] and active CPE consumes 7 Watt. While considering the 7 h (10 pm to 5 am) sleeping habit of rural people, the data plane devices can be entered into sleep mode during idle situations and the corresponding links can be shut down.

Figures 4 and 5 show the pattern of power consumption by all switches and links over the period of 24 h. The outcome of this empirical analysis indicates that switches and access points (APs) frequently entered into sleep mode whenever there was no traffic entering into the device and links. This leads to the total energy consumption rarely reaching peak value, except in some moments which look like spikes in the graphs.

We simplified the results to total annual energy consumption based on the 24 h of power utilization pattern by the network nodes and links. We achieved an annual energy saving of 31.50% on nodes and 55.44% on links with SoDIP6 network deployment. Note that nodes and switches/routers are used interchangeably in this article. Per unit charge of electricity in Nepal is almost equal to one cent. An annual energy bill to be paid by ISPs that have a different number of switches and links in their end access networks with respect to the legacy IPv4 network and the SoDIP6 network as per assumptions provided in our experiment is estimated and the comparative chart is shown in Figure 6. Hence, an ISP with 400 switches and 1800 links at its access network can have annual cost savings of up to three million USD. Most of the ISPs and Telcos of Nepal use Cisco, ZTE, and Huawei routers in

their networks. For Nepal Telecom, with tentatively 230 Huawei-NE40E-X3A routers (has a full mode power consumption of 600 Watts) and 1150 links, an estimated annual OpEX savings of 143.95K USD can be achieved.

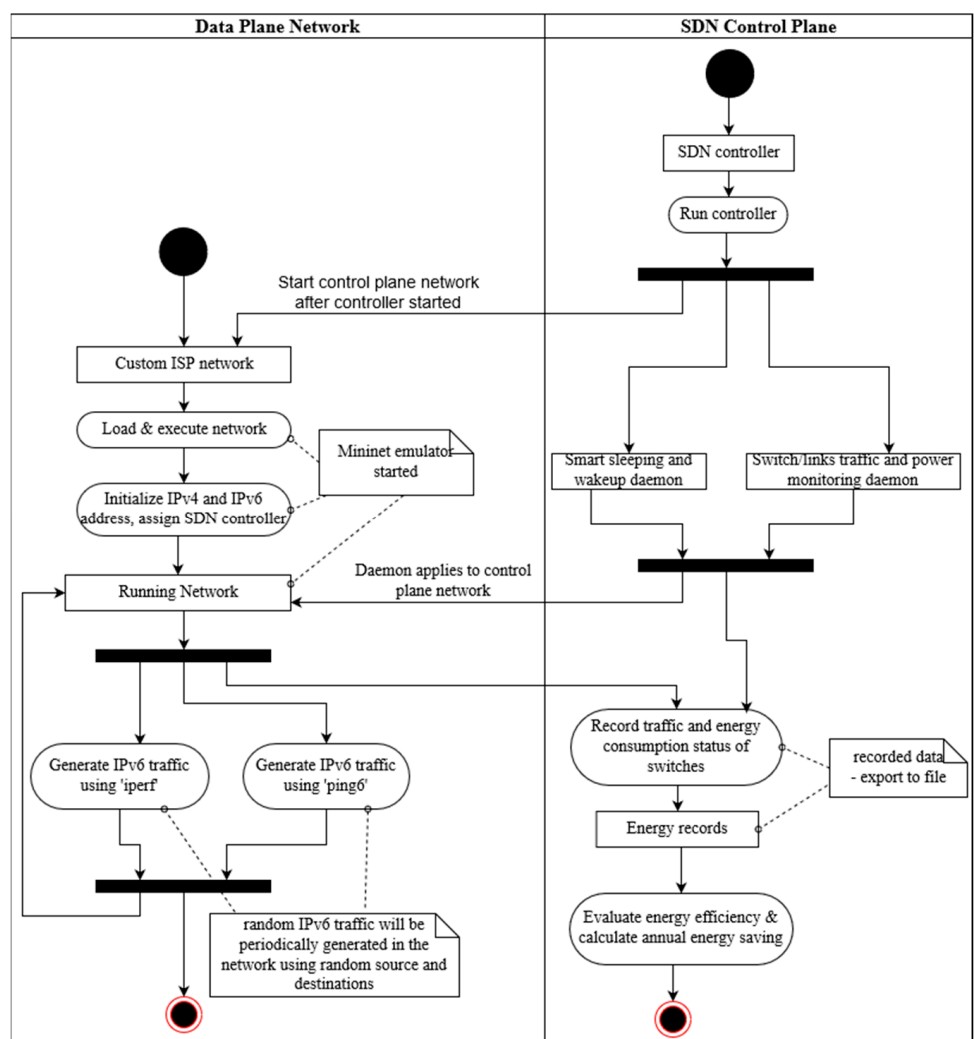

**Figure 3.** Activity diagram showing the steps at simulation environment. SDN, software defined networking.

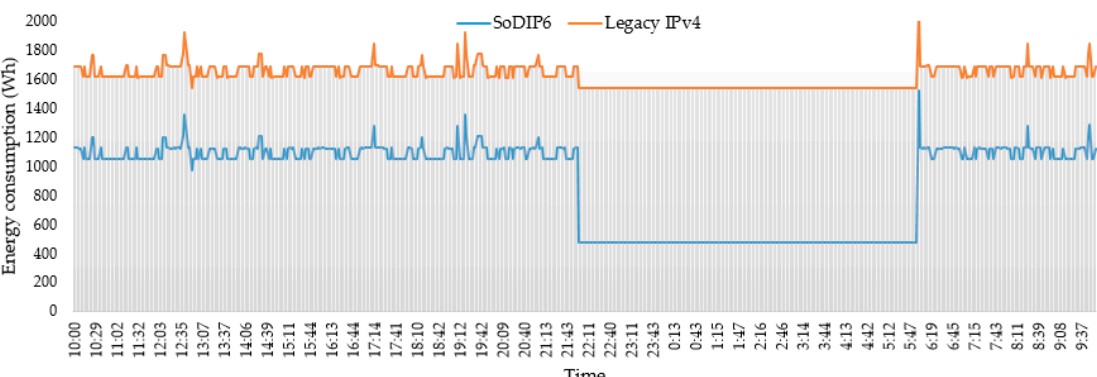

**Figure 4.** Pattern of total energy consumption by switches and customer premise equipment (CPE) in the legacy IPv4 and the software defined IPv6 (SoDIP6) network.

---

**Algorithm 1** Pattern of Energy Consumption Recording of the SoDIP6 Network

---

1.  */*(V, Ed) = G, where switches and CPEs as nodes (V) and links (Ed) in network topology graph G*/*
2.  *run_network(G)   /* initialize network, assign IPv6 address, set remote controller */*
3.  *src_list = random (V)   /* V belongs to CPEs. This identify source hosts for random ipv6 traffic generation.*/*
4.  *dst_list = random (V)/* V belongs to CPEs. This identify destination hosts to send IPv6 traffic generated form source hosts.*/*
5.  *while (running_network (G)):*
6.  *@every 2 min, Generate_IPv6_iperf(src_list)      /*generate IPv6 traffic every 2 min using iperf*/*
7.  *@every 3 min, Generate_IPv6_ping6(src_list)       /*generate IPv6 traffic every 3 min using ping6*/*
8.  *@every 2 to 3 min,         /*record the power of nodes and links*/*
9.  *For node in V:         /*record power of active and sleep node on every 2–3 min*/*
10. *if (node is active):*
11. *record_active_power(node)/*record active power of each switch with average traffic volume passing through it.*/*
12. *else:*
13. *record idle/sleep mode power of node*
14. *if no_traffic(node):       /* if node is idle then the node is signaled to enter into sleep mode */*
15. *sleep_node(node)*
16. *@every 2 to 3 min,        /* record power of links*/*
17. *for link in Ed:       /*record power of active and sleep switch on every 2–3 min*/*
18. *if (link is active):*
19. *record_link_power(link)      /*record power of each link*/*
20. *@10 PM to 6 AM, sleep_all_node()   /* sleep the node during the night from 10 PM to 6 AM */*
21. *@6 AM, wakeup_all_node()        /* wakeup nodes if in sleep mode */*

---

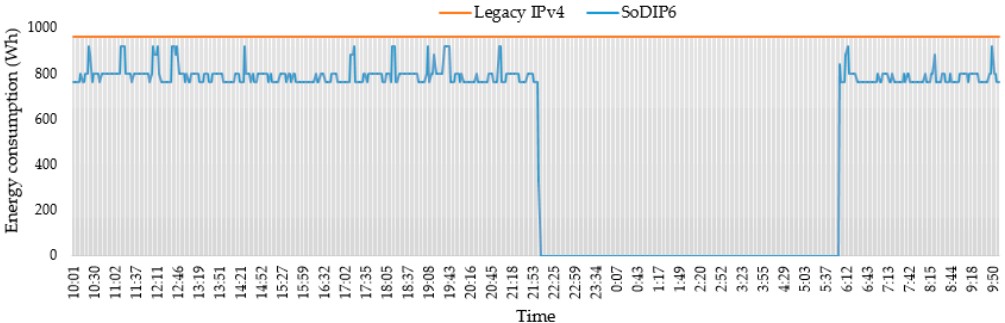

**Figure 5.** Pattern of total energy consumption by links in the legacy IPv4 and the SoDIP6 network.

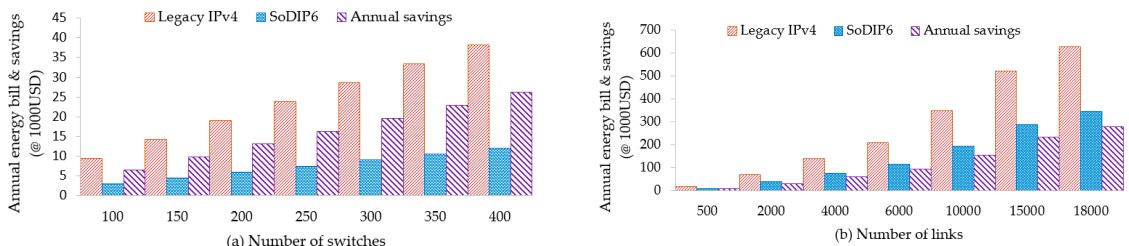

**Figure 6.** Comparative charts showing the estimated annual energy bill amount by both networks and operational expenditure (OpEX) savings in SoDIP6 network by (**a**) switches, and (**b**) links.

From the different literature [20,24–27] and our empirical analysis, it can be seen that SoDIP6 network is more energy efficient, leading to a reduction in organizational OpEX and hence contributes to community and service provider sustainability.

## 5. Conclusions

This paper presents unique challenges, solutions and recommendations for ICT expansion for developing country like Nepal. The unique geographical division of Nepal, ranging from high mountainous regions to flat Terai regions, enables as well as demands suitable localized research and innovation required to properly deploy appropriate broadband networks to make it affordable and sustainable, particularly to rural communities. We presented a study on different factors to realize affordable broadband services for rural communities and recommend solutions to make sustainable and affordable broadband services for rural communities. Our study recommends developing vision for suitable subsidization, provision for dedicated rural broadband policy, cost effective deployment of a wired rural network with newer technologies including SDN and IPv6, free content delivery to rural citizens, and many other solutions. Energy optimization in ICT networks has a large contribution to organizational OpEX savings, with a significant reduction in $CO_2$ emissions. Our experimental results show that the proposed SoDIP6 network helps ISPs save a huge amount on annual OpEX compared to that of legacy IPv4 based networks, while providing sustainable and affordable services to underserved rural communities.

**Author Contributions:** Conceptualization, B.R.D.; Methodology, B.R.D., D.S.B. and D.B.R.; Software, B.R.D.; Validation, B.R.D., D.B.R. and S.R.J.; Formal Analysis, B.R.D.; Investigation, B.R.D., D.B.R.; Resources, S.R.J.; Data Curation, B.R.D.; Writing—Original Draft Preparation, B.R.D.; Writing—Review & Editing, D.B.R. and S.R.J.; Visualization, B.R.D.; Supervision, D.B.R. and S.R.J.; Project Administration, B.R.D., D.S.B. and S.R.J.; Funding Acquisition, D.S.B. and S.R.J. All authors have read and agreed to the published version of the manuscript.

**Funding:** This research was funded by University Grant Commission-Nepal (FRG/74_75/Engg-1), Bhaktapur, Nepal.

**Acknowledgments:** We are thankful to following organizations for their supports in this research. (1) Center for Applied Research and Development (CARD), Institute of Engineering, Tribhuvan University, Kathmandu, Nepal, (2) Norwegian University of Science and Technology (NTNU), Trondheim, Norway, and (3) Nepal Academy of Science and Technology (NAST), Lalitpur, Nepal.

**Conflicts of Interest:** The authors declare no conflict of interest.

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
