# Peer review of "Affordable Broadband with Software Defined IPv6 Network for Developing Rural Communities"

_asi, doi:10.3390/asi3010004_

Round 1

Reviewer 1 Report

The paper is technically correct, but the main problem with this paper is that it is not clear WHAT IS NEW from the authors’ paper in [5].

It seems that all the ideas have appeared there and here a new scientific content is absent.

Even the result values of experiments with SoDIP6 network are identical to results in [5].

It seems as an autoplagiarism.

[5] B. R. Dawadi, D. B. Rawat, S. R. Joshi, and M. M. Keitsch, “Towards energy efficiency and 396 green network infrastructure deployment in Nepal using software defined IPv6 network 397 paradigm,” Electron. J. Inf. Syst. Dev. Ctries., 2019.

Author Response

We are thankful to reviewer for this wonderful comments that made us to clarify our contribution with this article.

we have attached the response of each points in a PDF file attached.

regards

Author(s)

Reviewer 2 Report

Content-related issues:

In sect. 3.3, it is suggested that the application of SDN and iPv6 would decrease the energy consumption in the network. However, no significant proof of this claim is presented. Line 199: the new acronym GHG is used with no explanation. 221 and l. 235: A submarine cable is mentioned, probably as a type of optical cable. I am afraid that this is not a suitable indication of the cable type, not only regarding its expected usage for underground installation. 327 mistyped switch name (“D5” instead of “E5”), also AP names (“A5” instead of “A3”) and cpes (“cpe0” instead of “cpe1”) Chapter 4, 3rd item in itemized list (QoS): It seems unclear whether there are any real implementation allowing to existing Flow-ID IPv6 header field so as the QoS improvement is achieved (citation needed). Chapter 4, last item in itemized list (Security): The claims here, namely “The Built-in IPSec framework in IPv6 implementation can significantly improve the network security” and “… Programmability features in SDN enables to apply different security models to avoid modern security breaches”) needs to be proven (e.g. by a citation. 331: The meaning of “operable” (ISP backbone) is unclear (e.g. IPv4 traffic is supported?) The acronyms CLAT and PLAX (line 331, 332, respectively) are used only once in the text, are they necessary? The Fig. 3 seems a bit unclear as a list of steps. I would recommend the usage of a standardized UML diagram. It is unclear from where the values for simulation (e.g. 0.01 Watt per Mbps at a switch) are taken from (citation!). End of Chapter 4. Missing comparison to non-SDN IPv4 network (one can suggest that a similar savings can be achieved using better switches without SDN as well).

Formal comments:

2: mistypes in Legend (“Interent” 2x)

Relatively numerous mistakes in English grammar, namely:

Missing articles in many places, Sometimes missing commas in sentences (e.g. after However in the sentence beginning). Other errors and mistypes (e.g. l. 127 “… people form the … zones “, l. 136: “… NTA initiations … does not …”), Wrong word order (e.g. l 152 “below sub sections.”)   Usage of uncommon phrases (e.g. “Like for example …”, “Sustainable issue” l. 285) Unusual word order, e.g. line 326

Author Response

Dear Reviewer,

We are heartily thankful for your constructive comments that reshape our article refined into standard quality.

a separate PDF file responding to each comments are attached here with.

additionally we have provided another pdf for reference to reviewer to make clear about our claim on "use of Submarine cable" for rural wired networks. Please just to note that this reference provided is not for publication.

Thank you and regards

Author(s)

Round 2

Reviewer 1 Report

The manuscript was improved and it can be accepted for the publication.